# Asymptomatic Idiopathic Renal Infarction Detected Incidentally on Contrast-Enhanced Computed Tomography: A Case Report

**DOI:** 10.3390/medicina59061176

**Published:** 2023-06-20

**Authors:** Mariko Mizusugi, Tsuneaki Kenzaka

**Affiliations:** 1Department of Internal Medicine, Hyogo Prefectural Tamba Medical Center, Tamba 669-3495, Hyogo, Japan; tag19fatk.3421@gmail.com; 2Division of Community Medicine and Career Development, Kobe University Graduate School of Medicine, Kobe 650-0017, Hyogo, Japan

**Keywords:** renal infarction, asymptomatic, rivaroxaban, normal findings, case report

## Abstract

*Background*: Renal infarction is an extremely rare disease. Although more than 95% of cases are symptomatic, there have been no previously reported asymptomatic cases, without any abnormal blood and urine test findings. Furthermore, the efficacy of long-term treatment of idiopathic renal infarction remains unknown. *Case Presentation:* A 63-year-old Japanese male underwent laparoscopy; a very low anterior resection of the rectum for lower rectal cancer (stage II) four years and five months prior to diagnosis with renal infarction. During the follow-up imaging studies, asymptomatic idiopathic renal infarction was found incidentally. The blood and urine test findings were normal. Contrast-enhanced computed tomography revealed a linearly bordered area of poor contrast in the dorsal region of the right kidney; however, no renal artery lesions, thromboembolic disease, or coagulation abnormalities were observed. Initial treatment with rivaroxaban 15 mg/day resulted in the remission of the infarcted lesion. The anticoagulation therapy was terminated after about 18 months without any incidences of re-infarction or bleeding events. *Conclusions:* We reported a very rare case of asymptomatic idiopathic renal infarction where blood and urine tests revealed no abnormal findings, and it was diagnosed incidentally during a post-treatment follow-up examination for lower rectal cancer. Long-term anticoagulant therapy for idiopathic renal infarction should be terminated at an appropriate time, taking the risk of bleeding into account.

## 1. Introduction

Renal infarction is an extremely rare condition, estimated to account for only 0.004–0.007% of emergency department visits [1]. More than 95% of cases are symptomatic, while over 90% present with abnormal blood or urine test findings such as elevated lactate dehydrogenase (LDH), high C-reactive protein (CRP), or microscopic hematuria, irrespective of the underlying diseases that contribute to renal infarction [2]. Underlying diseases include embolic conditions (including atrial fibrillation), renal artery involvement, and thrombotic conditions, but 3.8–30% are diagnosed as idiopathic cases without any underlying disease [1,2]. Furthermore, the treatment of renal infarction generally comprises six months of anticoagulant therapy in the initial phase of treatment, regardless of the disease background, but there is no standard treatment guideline or timeframe for long-term treatment of idiopathic renal infarction [3]. Therefore, asymptomatic renal infarction with no abnormalities in the blood and urine tests are quite rare, and idiopathic renal infarction is uncommon. Here, we report an unusual case of asymptomatic idiopathic renal infarction without any abnormal blood and urine tests and discuss its treatment modality.

## 2. Case Presentation

The patient was a 63-year-old Japanese male with no previous history of bleeding, infarction, or renal disease. The patient had a history of laparoscopic ultra-low anterior resection of the rectum for lower rectal cancer (stage II), followed by postoperative chemotherapy, which was four years and five months prior to the diagnosis of renal infarction. He underwent regular monitoring for cancer recurrence. In January 2021, at one such follow-up appointment, contrast-enhanced CT of the thorax and abdomen (Figure 1) incidentally revealed a linearly bordered area of contrast failure centered on the dorsal region of the right kidney, and the patient was referred to our department for consultation. He had no symptoms of abdominal or back pain, fever, nausea, or vomiting during the course of the cancer, and the CT findings of renal infarction were incidental. The patient had no relevant history other than lower rectal cancer. He was not taking any medications and had no history of smoking. He had no medical history of suspected COVID-19 and had not been vaccinated against coronavirus.

Examination revealed clear consciousness, blood pressure 163/113 mmHg, pulse 74 beats/min, temperature 36.7 °C, and respiratory rate 15 breaths/min. No bloody spots were observed on the eyelid conjunctiva. His heartbeat was regular, with no abnormal sounds, clear breath, and no difference between right and left pumping. His abdomen was flat, soft, and had no tenderness. He did not experience bilateral banging pain in the bilateral rib vertebral angles or skin rash. Laboratory test results were all within the normal range, as depicted in Table 1. Electrocardiogram (ECG) was in the normal sinus rhythm, and Holter ECG did not record any atrial fibrillation. There were no findings suggestive of local recurrence or metastasis of rectal cancer.

A diagnosis of right renal infarction was made based on the contrast-enhanced CT findings. Following the diagnosis, we investigated the background of the disease by conducting various examinations. Echocardiography revealed no verrucous or mucous species, left atrial enlargement, or shunt diseases, such as atrial septal defects. Both sets of blood culture tests were negative. As such, embolic renal infarction due to atrial fibrillation, infective endocarditis, or shunt disease was also ruled out. Renal magnetic resonance imaging and magnetic resonance angiography revealed no progressive atherosclerosis in the infarcted area, no stenosis or dysplasia of the renal arteries, and no aneurysm or arterial dissection (Figure 2). The patient was hypertensive at the time of admission, but a simple CT of the abdomen showed no calcification of the arteries. Therefore, renal infarction due to renal artery or vascular lesions caused by untreated hypertension was ruled out. Furthermore, tumor markers did not increase during the disease, and lower gastrointestinal endoscopy revealed no tumor lesions or findings suggestive of recurrence; therefore, a tendency to thrombosis due to recurrent rectal cancer was ruled out. Upper gastrointestinal endoscopy similarly showed no specific abnormalities. Blood tests were performed to screen for thrombophilia. Anti-nuclear antibodies, lupus anticoagulant, anti-CL-B2GP1 antibody, anti-cardiolipin IgG and IgM, anti-glomerular basement membrane antibody, proteinase-3-anti-neutrophil cytoplasmic antibodies (PR3-ANCA), and myeloperoxidase-anti-neutrophil cytoplasmic antibodies (MPO-ANCA) were all negative, while protein C activity, protein S activity, and AT-III were normal. In the absence of any signs of infection, a screening test for COVID-19 was not conducted. Overall, there was no evidence of any risk factors for renal infarction, including atrial fibrillation, renal artery injury, thrombophilia, and COVID-19 [4]. Hence, no background disease that could have caused coagulation abnormalities was detected; therefore, we diagnosed it as an idiopathic right renal infarction.

The patient was administered 15 mg/day of rivaroxaban for the treatment of atrial fibrillation, in accordance with the JCS/JHRS 2020 Guideline on Pharmacotherapy of Cardiac Arrhythmias [5]. Contrast-enhanced CT of the abdomen performed 7 months after the initiation of treatment showed that the contrast-impaired area of the right kidney had improved (Figure 3). Anticoagulant therapy mildly prolonged the prothrombin time-international normalized ratio and activated partial thromboplastin time but did not cause any bleeding events. There were no changes in other hematological or urinary findings, including renal function. There were no new episodes of atrial fibrillation. Contrast-enhanced CT of the abdomen performed approximately one year after the initiation of treatment showed no recurrent right renal infarction and no other new infarcts. Anticoagulant therapy was subsequently terminated after about 18 months. No embolic symptoms were observed 15 months post-treatment for renal infarction.

## 3. Discussion

To the best of our knowledge, this is the first case report of asymptomatic idiopathic renal infarction without any abnormal findings on blood and urine tests. This case was discovered incidentally during a follow-up of a malignant tumor and is an extremely rare case of renal infarction, discovered long after the onset of the disease without any subjective symptoms.

The frequency of symptoms and laboratory abnormalities of renal infarction was previously reported by Bourgault et al. [2]. Abdominal pain was reported in 50.9%, flank pain in 48.9%, nausea in 27.6%, vomiting in 20.2%, fever in 20.2%, hypertension in 48.0%, high LDH concentration in 90.5%, increase in CRP levels in 77.6%, high leukocyte count in 72.3%, impairment of kidney function in 40.4%, microscopic hematuria in 42.4%, and macroscopic hematuria in 5.9% of patients with renal infarction. Furthermore, 96.8% of patients reported abdominal or lateral abdominal pain as a symptom of initial renal infarction [2]. The frequency of symptom onset did not significantly differ with the disease background, which suggests that asymptomatic renal infarction is rare regardless of the underlying disease. White blood cell count and CRP levels that decreased over time almost returned to normal after 15 days of onset. The period of elevated LDH was the longest, and LDH may exceed the upper limit of normal even 15 days after onset [2]. In the present case, the patient showed no abnormalities in the blood tests, which indicates that LDH may have already peaked as it was assessed more than 15 days after disease onset.

The relevance of long-term treatment is also discussed in this study. There are currently no established treatment strategies in terms of duration of treatment for idiopathic renal infarction [3]. Table 2 depicts the reports obtained by searching for “idiopathic renal infarction” in PubMed [6,7,8,9,10,11,12,13,14]. Nine cases were published between 1995 and 2021. All patients were treated with anticoagulation and showed symptomatic improvement. Heparin and warfarin were used as anticoagulant therapy in most cases. In our present case, direct oral anticoagulant rivaroxaban improved the infarction, suggesting that the initial treatment contributes to the improvement of renal infarction. Moreover, the efficacy of long-term anticoagulant therapy in idiopathic renal infarction is controversial. The recurrence rate of idiopathic renal infarction is said to be low. In a one-year follow-up study of patients with idiopathic renal infarction by Oh et al. [15], no significant differences were observed in recurrence rates or complications associated with or without the use of anticoagulants. Alejandra García-García et al. previously reported that with a mean follow-up of 3.1 (±2.8) years, the rate of recurrent thrombotic events, such as arterial emboli and cerebrovascular disease was significantly lower in idiopathic renal infarction (0%) than in secondary renal infarction (18.8%) [16]. Conversely, in a study investigating the significance of long-term anticoagulant therapy, Khayat et al. [3] showed that the event rates after a median follow-up of 4.3 years in 103 patients with idiopathic renal infarction were comparable between the anticoagulation and non-anticoagulant groups, including those administered antiplatelet drugs. Although there was no significant difference in the incidence of thromboembolic events between the anticoagulant and non-anticoagulant groups, bleeding events were significantly higher in the anticoagulant group (13%). Because of this, long-term anticoagulation in idiopathic renal infarction with a low recurrence rate may only increase the risk of bleeding. Therefore, discontinuing anticoagulation therapy after initial treatment may be an ideal option. However, there are no reports on the incidence of thrombotic events in renal infarction after the end of anticoagulant therapy, and the appropriate duration of anticoagulant therapy thus remains unclear. Therefore, the appropriate duration of anticoagulant therapy, needs to be investigated through future case series.

## 4. Conclusions

In this case, we reported a very rare case of asymptomatic idiopathic renal infarction without any abnormal findings in blood and urine tests. The diagnosis was made incidentally during a post-treatment follow-up for lower rectal cancer. Regarding the treatment strategy for renal infarction, long-term anticoagulant therapy should be terminated at an appropriate time, taking the risk of bleeding into account.

## Figures and Tables

**Figure 1 medicina-59-01176-f001:**
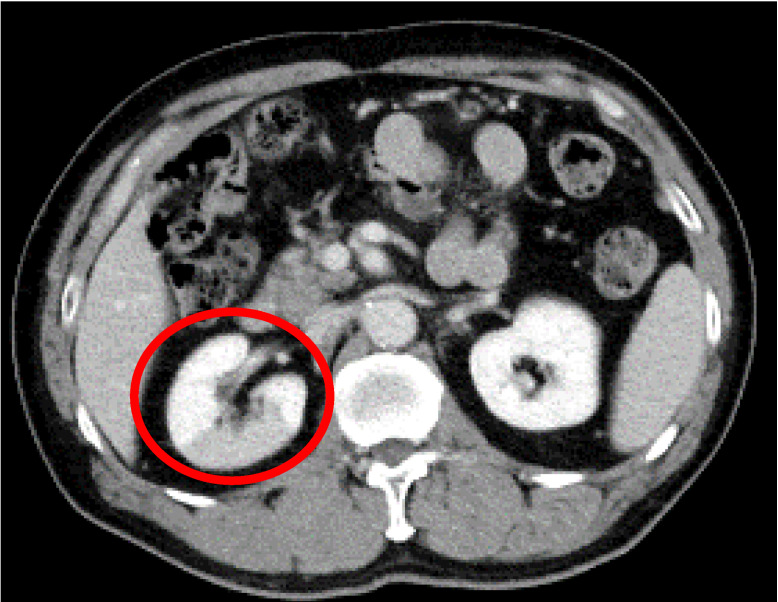
Contrast-enhanced computed tomography image of the abdomen. A contrast-impaired area with a straight border centered on the dorsal region of the right kidney is observed (red circle).

**Figure 2 medicina-59-01176-f002:**
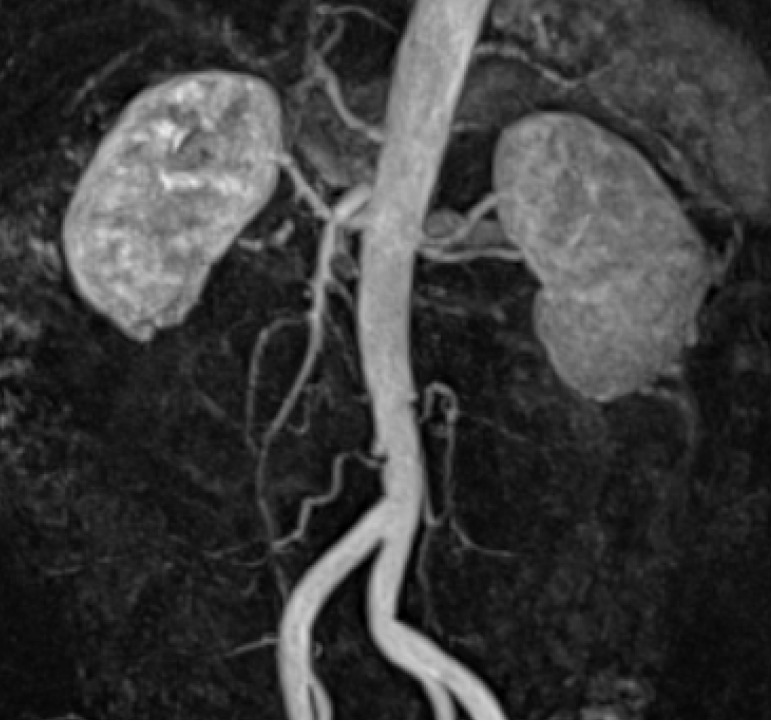
Renal magnetic resonance angiography. No progressive atherosclerosis, stenosis, or dysplasia of the renal artery is observed in the infarcted area. No aneurysm or arterial dissection occurred.

**Figure 3 medicina-59-01176-f003:**
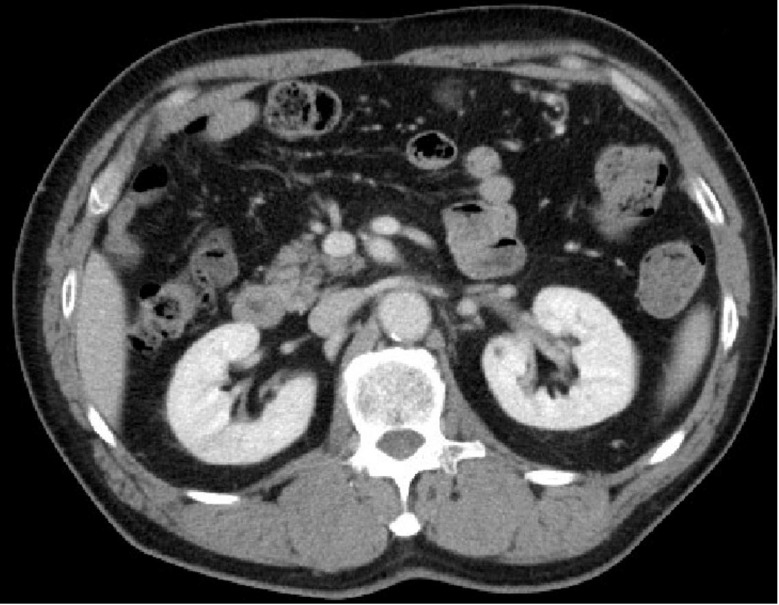
Contrast-enhanced computed tomography image of the abdomen. The contrast-impaired area of the right kidney had improved seven months after the initiation of rivaroxaban.

**Table 1 medicina-59-01176-t001:** Laboratory data upon admission.

Parameter	Recorded Value	Standard Value
White blood cell count	8090 µL	4500–7500 µL
Hemoglobin	15.0 g/dL	11.3–15.2 g/dL
Platelet count	43.6 × 10^4^ µL	13–35 × 10^4^ µL
Prothrombin time/International normalized ratio	0.84	0.80–1.20
Activated partial thromboplastin time	27.5 s	26.9–38.1 s
D-dimer	1.0 μg/mL	<1.0 μg/mL
C-reactive protein	0.13 mg/L	≤0.60 mg/dL
Aspartate aminotransferase	21 U/L	11–30 U/L
Alanine aminotransferase	20 U/L	4–30 U/L
Lactase dehydrogenase	181 U/L	109–216 U/L
Creatine kinase	73 U/L	40–150 U/L
Blood urea nitrogen	9.5 mg/dL	8–20 mg/dL
Creatinine	0.95 mg/dL	0.63–1.03 mg/dL
Sodium	139 mEq/L	136–148 mEq/L
Potassium	4.1 mEq/L	3.6–5.0 mEq/L
Chloride	102 mEq/L	98–108 mEq/L
CEA	1.94 ng/mL	<5.0 ng/mL
CA19-9	10.4 U/mL	<37.0 U/mL

CEA: Carcinoembryonic antigen, CA19-9: Carbohydrate antigen 19-9.

**Table 2 medicina-59-01176-t002:** The published case reports of idiopathic renal infarction. NR means “it wasn’t reported in the paper”.

Case	Literature	Year Published	Age (Years)	Sex	Symptoms	Treatment	Long-Term Outcomes after Treatment
1	6	2017	38	Male	Right-sided abdominal pain, Radiating pain to the iliac fossa and back	Continuous intravenous enoxaparin sodium was started; it was switched to oral warfarin.	No recurrence after one year.
2	7	2021	43	Male	Right renal colic	After intravenous heparin administration, the medication was switched to vitamin K antagonists.	No recurrence at 3, 6, and 12 months post-treatment.
3	8	2018	41	Male	Right upper and lateral abdominal pain, Fever and chills	Therapeutic dose of enoxaparin sodium	NR
4	9	2017	42	Male	Nausea, Epigastric pain, Radiating pain to the back	Oral anticoagulant (type and dose of the medicine were not reported)	NR
5	10	2021	37	Male	Left-sided abdominal colic, Radiogenic pain to testis	Unfractionated heparin 80 mg/kg bolus and continuous intravenous infusion at 18 mg/kg/h. This was then switched to oral warfarin.	NR
6	11	2014	50	Male	Right-sided abdominal pain, Vomiting, Sweating, Loss of appetite	Enoxaparin sodium and warfarin administration. Lisinopril oral administration.	No recurrence after 6 months.
7	12	1995	37	Male	Lateral abdominal pain, Vomiting, Slight fever	After two weeks of intravenous heparin, six months of intravenous warfarin was administered.	No recurrence at 8 weeks post-CT.
8	13	2005	42	Female	Left-sided abdominal pain, Radiating pain to the left lower abdomen	Warfarin (drug dosage was not reported)	NR
9	14	2009	46	Male	Right-sided abdominal pain, Radiating pain to the right lower abdomen	Systemic anticoagulation after selective transarterial urokinase infusion (type and dose of the medicine were not reported)	NR

## Data Availability

All data generated or analyzed during this study are included in the published article.

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
