# Peer review of "Asymptomatic Idiopathic Renal Infarction Detected Incidentally on Contrast-Enhanced Computed Tomography: A Case Report"

_medicina, 2023, doi:10.3390/medicina59061176_

Round 1

Reviewer 1 Report

To whom it may concern

The renal infarction case presentation is relevant and relatively well documented. The article includes a clear and concise abstract. The introduction is describing etiological heterogenicity of renal infarction.

General comments:

1. please specify if the patient was diagnosed with RI during COVID pandemics. In this situation, I would add if the patient was screened for recent COVID, titer of antibodies (if determined) and if he was vaccinated or not.

2. Please provide some more information about risk factors, like COVID, I suggest the authors several articles, like:

-W. Miesbach, M. Makris.COVID-19: Coagulopathy, risk of thrombosis and the rationale for anticoagulation. Clin Appl Thromb/Haemost, 26 (2020), pp. 1-7

-  N.P. Murray, C Fuentealba, E Reyes, A Salazar. Renal infarction associated with asymptomatic Covid-19 infection. Hematol Transfus Cell Ther. 2021;43(3):353−356

5. Please provide information about the guidelines you used to treat your patient and asymptomatic RI, with no clotting disorder or C-V abnormality for 18 months with NOAC!!

Author Response

The renal infarction case presentation is relevant and relatively well documented. The article includes a clear and concise abstract. The introduction is describing etiological heterogenicity of renal infarction.

Response: Thank you for the careful review of our manuscript and for sharing your valuable feedback.

  1. Please specify if the patient was diagnosed with RI during COVID pandemics. In this situation, I would add if the patient was screened for recent COVID, titer of antibodies (if determined) and if he was vaccinated or not.

Response: Thank you for your suggestion. The patient was diagnosed with renal infarction in January 2021. Subsequently, the patient had no medical history of suspected COVID-19, so we did not conduct a screening test. Following your suggestion, we have added the sentence pertaining to COVID -19 (Lines 58-59). The following sentence was added: “He had no medical history of suspected COVID-19 and had not been vaccinated against coronavirus.”

  1. Please provide some more information about risk factors, like COVID, I suggest the authors several articles, like:

-W. Miesbach, M. Makris.COVID-19: Coagulopathy, risk of thrombosis and the rationale for anticoagulation. Clin Appl Thromb/Haemost, 26 (2020), pp. 1-7

-  N.P. Murray, C Fuentealba, E Reyes, A Salazar. Renal infarction associated with asymptomatic Covid-19 infection. Hematol Transfus Cell Ther. 2021;43(3):353−356

Response: Thank you for your advice. We did not conduct a screening test for COVID-19. Hence, we added this information to Line 109. The added sentence is “In the absence of any signs of infection, a screening test for COVID-19 was not conducted.” Following this sentence, to highlight some risk factors for renal infarction and add more clarity, we added this sentence; “Overall, there was no evidence of any risk factors for renal infarction including atrial fibrillation, renal artery injury, thrombophilia, and COVID-19 [4]."

  1. Please provide information about the guidelinesyou used to treat your patient and asymptomatic RI, with no clotting disorder or C-V abnormality for 18 months with NOAC!!

Response: Thank you for your valuable suggestion. We referred to the treatment for atrial fibrillation as described in the JCS/JHRS 2020 Guideline on Pharmacotherapy of Cardiac Arrhythmias. We have added the pertinent information to Line 101-103. The sentence was modified to, “The patient was administered 15 mg/day of rivaroxaban for the treatment of atrial fibrillation, in accordance with the JCS/JHRS 2020 Guideline on Pharmacotherapy of Cardiac Arrhythmias.”

Furthermore, in the discussion section, we noted that at one year, there was no significant difference in the recurrence rate or complications associated with or without anticoagulant use; however, at four years, there was a marked increase in complications. For this reason, we used anticoagulants for only a little over one year and subsequently discontinued them.

Reviewer 2 Report

The authors reported an unusual case of asymptomatic idiopathic renal infarction without any abnormal blood and urine tests, and discussed its treatment modality. Based on this, the authors concluded that long-term anticoagulant therapy for idiopathic renal infarction should be terminated at an appropriate time, taking the risk of bleeding into account. I did not have any major concerns, only several minor issues listed below:

In Table 2, please correct the reference numbers. For instance, case 1 should refer to literature 4 instead of 5.

Author Response

The authors reported an unusual case of asymptomatic idiopathic renal infarction without any abnormal blood and urine tests and discussed its treatment modality. Based on this, the authors concluded that long-term anticoagulant therapy for idiopathic renal infarction should be terminated at an appropriate time, taking the risk of bleeding into account.

Response: Thank you for the careful review of our manuscript and for sharing your valuable feedback. We appreciate your encouraging comments.

In Table 2, please correct the reference numbers. For instance, case 1 should refer to literature 4 instead of 5.

Response: Thank you for pointing out this discrepancy. We have likewise corrected the reference numbers in Table 2.

Reviewer 3 Report

Thank you for the possibility to review the manuscript: “Asymptomatic idiopathic renal infarction detected incidentally on contrast-enhanced computed tomography: A case report”.

This condition is rare and it is difficult to reach the experience how to properly manage the consequences.

The introduction is informative.

Case report is well described.

In table 1 platelet number units are different for current and normal value.

The discussion reflects well the literature data of cases of patients with renal infarction with current patient.  Also it is important to discuss treatment procedures, what authors have done.

Author Response

Thank you for the possibility to review the manuscript: “Asymptomatic idiopathic renal infarction detected incidentally on contrast-enhanced computed tomography: A case report”.

This condition is rare and it is difficult to reach the experience how to properly manage the consequences.

The introduction is informative.

Case report is well described.

In table 1 platelet number units are different for current and normal value.

The discussion reflects well the literature data of cases of patients with renal infarction with current patient.  Also it is important to discuss treatment procedures, what authors have done.

Response: Thank you for your careful review and your encouraging comments. Regarding the platelet counts, they did not increase or decrease and were not considered to be of pathological significance. The standard value was revised from 13–35 × 103/µL to 13–35 × 104/µL.
